# Automated Radiosynthesis, Preliminary In Vitro/In Vivo Characterization of OncoFAP-Based Radiopharmaceuticals for Cancer Imaging and Therapy

**DOI:** 10.3390/ph15080958

**Published:** 2022-08-02

**Authors:** Francesco Bartoli, Philip Elsinga, Luiza Reali Nazario, Aureliano Zana, Andrea Galbiati, Jacopo Millul, Francesca Migliorini, Samuele Cazzamalli, Dario Neri, Riemer H. J. A. Slart, Paola Anna Erba

**Affiliations:** 1Nuclear Medicine, Department of Translational Research and Advanced Technologies in Medicine and Surgery, University of Pisa and Azienda Ospedaliero Universitaria Pisana, 56126 Pisa, Italy; francesco.bartoli@med.unipi.it; 2Medical Imaging Center, University Medical Center Groningen, University of Groningen, 9712 CP Groningen, The Netherlands; p.h.elsinga@umcg.nl (P.E.); l.reali.nazario@umcg.nl (L.R.N.); r.h.j.a.slart@umcg.nl (R.H.J.A.S.); 3Philochem AG, R&D Department, Libernstrasse 3, CH-8112 Otelfingen, Switzerland; aureliano.zana@philochem.ch (A.Z.); andrea.galbiati@philochem.ch (A.G.); jacopo.millul@philochem.ch (J.M.); francesca.migliorini@philochem.ch (F.M.); samuele.cazzamalli@philochem.ch (S.C.); 4Department of Chemistry and Applied Biosciences, Swiss Federal Institute of Technology, CH-8093 Zurich, Switzerland; dario.neri@philogen.com; 5Philogen S.p.A., 53100 Siena, Italy; 6Biomedical Photonic Imaging Group, Faculty of Science and Technology, University of Twente, 7522 NB Enschede, The Netherlands

**Keywords:** ^68^Ga-labeled, ^177^Lu-labeled, Al[^18^F]F-labeled, OncoFAP, automated radiosynthesis, positron emission tomography, radiopharmaceuticals, tumors imaging, theranostics

## Abstract

FAP-targeted radiopharmaceuticals represent a breakthrough in cancer imaging and a viable option for therapeutic applications. OncoFAP is an ultra-high-affinity ligand of FAP with a dissociation constant of 680 pM. OncoFAP has been recently discovered and clinically validated for PET imaging procedures in patients with solid malignancies. While more and more clinical validation is becoming available, the need for scalable and robust procedures for the preparation of this new class of radiopharmaceuticals continues to increase. In this article, we present the development of automated radiolabeling procedures for the preparation of OncoFAP-based radiopharmaceuticals for cancer imaging and therapy. A new series of [^68^Ga]Ga-OncoFAP, [^177^Lu]Lu-OncoFAP and [^18^F]AlF-OncoFAP was produced with high radiochemical yields. Chemical and biochemical characterization after radiolabeling confirmed its excellent stability, retention of high affinity for FAP and absence of radiolysis by-products. The in vivo biodistribution of [^18^F]AlF-NOTA-OncoFAP, a candidate for PET imaging procedures in patients, was assessed in mice bearing FAP-positive solid tumors. The product showed rapid accumulation in solid tumors, with an average of 6.6% ID/g one hour after systemic administration and excellent tumor-to-healthy organs ratio. We have developed simple, quick, safe and robust synthetic procedures for the preparation of theranostic OncoFAP-compounds based on Gallium-68, Lutetium-177 and Fluorine-18 using the commercially available FASTlab synthesis module.

## 1. Introduction

Small organic ligands that selectively bind to tumor-associated antigens are increasingly applied as targeting delivery vehicles of small bioactive payloads such as radionuclides [1,2]. Fibroblast activation protein (FAP, FAP-α) is a type-II transmembrane serine protease overexpressed on the surface of cancer-associated fibroblasts (CAFs) in the tumor microenvironment (TME) in most epithelial malignancies [3]. FAP has recently emerged as a novel in vivo imaging biomarker with prognostic and theranostic potential. Favorable biodistribution profile and tumors-to-background ratios have been demonstrated for several FAP-based radiopharmaceuticals in development for PET/CT imaging. A wide range of cancer types has been imaged successfully with FAP-radioligands based on small molecules, particularly in those clinical conditions where imaging with [^18^F]FDG, the most extensively used PET radiopharmaceutical in oncology, has encountered limitations [4,5,6,7,8].

To the best of our knowledge, OncoFAP is the FAP small organic ligand with the highest-affinity reported to date, with a dissociation constant of 680 pM compared with that of FAPI-04 of 1020 pM [9]. OncoFAP-radio and OncoFAP-fluorophore conjugates exhibit rapid, selective and efficient tumor-targeting performance in murine models of cancer [9]. Extensive pre-clinical characterization of [^68^Ga]Ga-DOTAGA-OncoFAP demonstrates its excellent imaging performance with similar targeting properties in a head-to-head comparison with [^68^Ga]Ga-FAPI-46 [8]. First-in-human data have recently confirmed the pan-tumoral targeting potential of [^68^Ga]Ga-DOTAGA-OncoFAP, with high uptake in primary tumors (breast cancer, colon cancer, fibrosarcoma, hepatocellular carcinoma), lymph node and distant metastases and a rapid clearance from healthy organs [8].

Given the high expression of FAP in cancer and the absence of significant uptake of FAP-targeting small molecule tracers in normal tissues, OncoFAP-based radioligands have inherent potential for theranostic applications. DOTAGA-OncoFAP can be radiolabeled with radioactive metals entailing therapeutic potential such as lutetium-177, the current most popular radionuclide for therapeutic applications for neuroendocrine tumors [10] and prostate cancer [11], leading to a new family of theranostic radiopharmaceuticals.

The interest of using DOTAGA for binding metal radionuclides is not limited to ^68^Ga-labeled imaging agents. Indeed, DOTAGA can be used to complex several other interesting radiometals, such as lutetium-177, scandium-44, indium-111, zirconium-89 and actinium-225 [12,13,14]. In this work, we present the chemical, biological and pre-clinical characterization of a series of radiotheranostics OncoFAP derivatives, specifically DOTAGA-OncoFAP, NODAGA-OncoFAP and NOTA-OncoFAP with gallium-68 and fluorine-18 for PET/CT imaging and with lutetium-177 for therapeutic applications. Further, the production of a simple preparation kit was also developed for [^68^Ga]Ga-DOTAGA-OncoFAP and [^68^Ga]Ga-NODAGA-OncoFAP radiotracers.

## 2. Results

OncoFAP-COOH was used as common precursor for the synthesis of DOTAGA-OncoFAP, NODAGA-OncoFAP and NOTA-OncoFAP (Figure 1). Final compounds were afforded by following standard coupling and deprotection procedures. Detailed synthetic procedures, chemical structures and product characterization are reported in the Appendix A.

### 2.1. Radiosynthesis

#### 2.1.1. [^68^Ga]Ga-OncoFAP-Derivatives

The production of [^68^Ga]Ga-DOTAGA-OncoFAP, [^68^Ga]Ga-NODAGA-OncoFAP and [^68^Ga]Ga-NOTA-OncoFAP using a 5–10 min automatic synthesis procedure at a reaction temperature of 95 °C was highly reproducible, with high radiochemical purity (RCP) and high radiochemical yields (RCY, Table 1, Table 2, Table 3 and Table 4). The molar activity was 20–30 GBq/μmol with excellent RCY, exceeding 80% for >15 μg precursors amount. The best RCY were obtained using 25 μg of OncoFAP-derivates, a reaction time of 5 min and a reaction pH of 4.2. No significant differences were observed increasing the amount of precursor to 20 µg and 25 µg or increasing the reaction time to 10 min. The validation data of our synthesis conditions are reported in Table 5.

For the OncoFAP kit, the best result was obtained with 10 min radiolabeling at a reaction temperature of 95 °C with 40 μg of precursor and pH of 3.2 (Appendix A). The reduction of the pH to 3 using formate buffer was necessary to reduce the colloidal gallium impurity below 1.5 ± 0.2%. In fact, the use of the same condition of the automatic synthesis procedure (acetate buffer at pH 4.2) resulted in RCP of 88 ± 2.4% with significant amount of both free gallium-68 and colloidal gallium-68 (Table 5 and Table 6).

#### 2.1.2. [^18^F]AlF-NOTA/NODAGA-OncoFAP

The radiosynthesis of [^18^F]AlF-NOTA-OncoFAP via [^18^F]AlF-approach resulted in a RCY of 20.9% with a RCP of 89.6% (Table 7). The synthesis of [^18^F]AlF-NODAGA-OncoFAP resulted in very low RCY and low RCP (Table 8). The best result for the NOTA-precursor was obtained using 250 μg of precursor with a reaction time of 25 min and a reaction temperature of 95 °C (Table 9).

#### 2.1.3. [^177^Lu]Lu-DOTAGA-OncoFAP

For the radiosynthesis of [^177^Lu]Lu-DOTAGA-OncoFAP, we obtained high molar activities of about 100 GBq/µmol with high RCY and RCP > 99.8% (Table 10 and Table 11) using a reaction time of 30 min and a reaction temperature of 95 °C at a pH 4. We validated the radiolabeling method repeating the synthesis 3 times at the highest molar activity, simulating a typical production of a batch for PRRT i.e., 9.2 GBq of Lutetium-177 (Table 11). The RCY of [^177^Lu]Lu-DOTAGA-OncoFAP strictly depends on the reaction volume and on the amount of precursor used in synthesis. The preparation was easily purified through C18 cartridge to remove the excess free Lutetium-177. No loss of [^177^Lu]Lu-DOTAGA-OncoFAP was observed in this process (Appendix A).

### 2.2. Preliminary In Vitro/In Vivo Characterization of OncoFA-Derivatives

The co-elution experiments showed that all the OncoFAP derivatives described in this work retained the ability to form stable complexes with recombinant hFAP (Figure 2 and Appendix A).

All [^68^Ga]Ga-OncoFAP and [^18^F]AlF-OncoFAP radio-conjugates were highly stable (>99% of intact compound) in 0.9% saline solution and in human plasma at 37 °C for 2 h. On the other hand, a stability test of [^177^Lu]Lu-DOTAGA-OncoFAP radiolabeled batch showed the presence of radiolysis of about 10% every 24 h, which was significantly reduced to about 2% within 8 days by the addition of 20 mg of gentisic acid as a radical scavenger (Figure 3b) [15,16]. All the radiolabeled preparations of OncoFAP derivatives exhibited high hydrophilicity. Details are given in Appendix A.

Cell binding studies showed a similar trend of binding on FAP-positive SK-RC-52 cells for all the OncoFAP radio-conjugates. All the compounds displayed an intense binding at early time point (10 min), followed by a progressive decrease over time. As shown in Figure 4, a major difference in the absolute binding value was found among 20% for [^177^Lu]Lu-DOTAGA-OncoFAP, 2.3% for [^68^Ga]Ga-DOTAGA-OncoFAP and 0.25% for [^18^F]AlF-NOTA-OncoFAP. Results of the nonspecific binding on SK-R-RC-52 wild-type (SK-RC-52.wt) showed a very low absolute binding value for all the compounds (Figure 4).

In vivo biodistribution of [^18^F]AlF-NOTA-OncoFAP (500 nmol/kg, ~4 MBq/Kg, RCP 89%) in athymic Balb/c AnNRj-Foxn1 mice bearing subcutaneous HT-1080.hFAP fibrosarcoma showed selective accumulation in FAP-positive tumors (6.6% ID/g, 1 h post intravenous injection), with excellent selectivity against healthy organs. Tumor-to-blood and tumor-to-kidney ratios of 6.5-to-1 and 4.3-to-1, respectively, were observed at the 1-h time point (Figure 5).

## 3. Discussion

In recent years, the development of tumor-targeting ligands resulted in the discovery of a new generation of diagnostic and therapeutic products with high uptake in cancer lesions and low accumulation in healthy organs. Lutathera^®^ and [^177^Lu]Lu-PSMA-617 provide examples of radioligand therapeutics with proven clinical efficacy and limited systemic toxicity [17,18]. Radioligand diagnostic and therapeutic products targeting FAP in the tumor microenvironment represent a promising class of compounds with pan-tumoral applicability and high selectivity for tumor lesions [19,20].

FAP-targeting strategies have gained growing relevance in nuclear medicine for the development of radiolabeling. Although anti-FAP antibodies have been known since the 1990s [21,22,23,24], the discovery of small organic FAP ligands in the last few years represented a revolution in the field of nuclear medicine. Various FAP-targeting agents coupled to different radionuclide chelators have been recently developed and characterized for their high affinity and selectivity towards FAP-positive tumors. These conjugates exhibit rapid accumulation in cancer lesions and low uptake in healthy organs both in preclinical murine models and cancer patients. Among FAP-specific small organic ligands, OncoFAP is the compound with the highest affinity described so far (Kd = 680 pM) [9]. Preliminary data with [^68^Ga]Ga-DOTAGA-OncoFAP have demonstrated the feasibility of ^68^Ga-radiolabeling and highly favorable targeting properties in both small animal and patients with cancer, validating [^68^Ga]Ga-OncoFAP as a new powerful alternative to clinically established PET tracers [8]. BiOncoFAP, a bivalent derivative of OncoFAP, has been recently described as a novel compound with high and prolonged tumor uptake in murine models of cancer [25]. With the aim to identify a clinical [^18^F]-OncoFAP candidate, we generated NODAGA-OncoFAP and NOTA-OncoFAP as two preclinical prototypes that could be labeled both with Fluorine-18 and with Gallium-68. NODAGA-OncoFAP rapidly complexes Gallium-68 with excellent product stability, thus opening promising avenues to the development of simple kit preparations

The translation of radiopharmaceutical agents from preclinical development to clinical applications requires the implementation of standard synthetic and radiolabeling procedures [26]. In order to facilitate clinical development of novel FAP-radiopharmaceuticals, we have developed highly reproducible automated methodologies for the efficient radiolabeling of OncoFAP-derivatives (DOTAGA-OncoFAP, NODAGA-OncoFAP and NOTA-OncoFAP) with Gallium-68, Fluorine-18 and Lutetium-177 radioisotopes. The procedures described in this article are easy to implement, safe and robust. We based these procedures on the FASTLab automated module, a radiolabeling platform already broadly applied to produce other radiopharmaceuticals at the commercial scale [27,28,29]. With the aim to enable production of [^68^Ga]-DOTAGA/NODAGA-OncoFAP radiopharmaceuticals also in small radiopharmacy sites with limited financial resources, we have also generated a radiolabeling single-vial cold kit. This approach, already proposed for other products (Illuccix™ and NETSPOT^®^; SOMAKIT TOC) [30,31,32], renders the production of ^68^Ga-based products as easy as the one of ^99m^Tc-based radiopharmaceuticals. The “OncoFAP-kit”, which already contains all necessary items (buffers, ligands and excipients) to be combined with the radioisotopes, allows efficient production of [^68^Ga]Ga-NODAGA-OncoFAP and [^68^Ga]Ga-DOTAGA-OncoFAP for PET imaging applications. The reasons for the development of such a strategy are related to the need to simplify and facilitate, while maintaining high standard of quality, efficiency and reproducibility the preparation of ^68^Ga imaging agents. Indeed, the traditional approach of automatic synthesis of ^68^Ga-based radiopharmaceuticals generates substantial investments, i.e., hot cells and synthesis modules, quality control equipment, and highly qualified personnel who may not be easily accessible to small-scale radiopharmacies, thus hampering the widespread use of ^68^Ga-PET imaging agents and its equal accessibility for patients throughout the world. Consequently, interest has grown on the development of cold kits for PET tracers. The main advantages of a kit-based process are lower investments, ease of use, absence of purification steps (only a 0.22 µm sterilizing filter is required at the generator outlet to ensure the sterility of the final formulation), absence of EtOH and the lower final volume at the cost of higher amounts of precursor required, thus resulting in lower molar activity. Further, the lack of purification requires high-quality radioactive eluate.

Very few data are currently available on the direct comparisons of the operator’s radiation exposure when producing ^68^Ga-radiopharmaceuticals using on automatic synthesis modules and simple kits. Frinde et al., by measuring the dose at the extremities via proximity meters fixed to the first phalanx of each middle finger for a 1.85 GBq generator at the date of calibration, reported 70 μSv for the left hand and 132 μSv for the right hand for labeling with an automatic synthesis module but 179 μSv and 152 μSv for kit-based preparation of [^68^Ga]Ga-DOTANOC [33], The measurements made by Kleynhans et al. during the synthesis of [^68^Ga]Ga-HBED-CC-PSMA-11 demonstrated a whole-body exposure significantly lower for automated synthesis with 2.05 ± 0.99 μSv versus 14.32 ± 5.3 μSv for the kit-based preparation [34] as a result of the different radiolabeling environment in which the procedures were performed: a hot cell with 50 mm lead shield for the automated synthesis as compared to a class II biosecurity cabinet shielded with a 3 mm lead layer, with a 10 mm lead shielded desktop screen for the kit-based preparation. When the same 50 mm lead hot cell was used, the whole-body exposure dropped to 2 ± 0.5 μSv also for the kit-based preparation [35] with a dose at the extremities of about 1.5 ± 0.4 mSv. Therefore, reducing the exposure for simple kit labeling is possible, but dependent on the environment of the radiolabeling and on the experience of the operator.

As an alternative to Gallium-68 for diagnostic applications, we also developed two novel OncoFAP-conjugates bearing NODAGA and NOTA radiometal chelators, which are known to form stable complexes with Alluminum-Fluorine-18 [36,37,38,39,40]. We performed the radiochemical synthesis of both [^18^F]AlF-NODAGA-OncoFAP and [^18^F]AlF-NOTA-OncoFAP. The radiolabeling of NODAGA-OncoFAP with Alluminum-Fluorine-18 resulted in very low RCY and RCP, probably due to the ability of the NODAGA chelator to form a stable neutral complex with the Al^3+^ ion, thus partially preventing the incorporation of the fluorine-18 as also reported by Liu et al. [41]. On the contrary, NOTA-OncoFAP, which devoid a carboxylic acid function, was efficiently radiolabeled with Aluminium-Fluorine-18, with a good purity and a RCY of approximately 20%. DOTAGA-OncoFAP can be labeled both with Gallium-68 and Lutetium-177, and thus can represent a “theranostic pair”. This approach has been already successfully implemented for the diagnosis and therapy of Neuro-Endocrine Tumors (Lutathera^®^/NETSPOT^®^) [10] and prostate cancers ([^68^Ga]Ga-PSMA-11/[^177^Lu]Lu-PSMA-617) [11]. Our data prove the feasibility of producing high-quality [^177^Lu]Lu-DOTAGA-OncoFAP with molar activities of 100 GBq/µmol, which is almost double the molar activities currently routinely achieved with [^177^Lu]Lu-PSMA-617 [42]. Incorporating the high activity of Lutetium-177 represents a critical parameter to obtain a therapeutic radiopharmaceutical (i.e., in the range of 7.4–14.8 GBq) containing relatively low amounts of precursor, thus avoiding saturation of targets at the tumor site [9]. Such results serve as the basis to provide a fast track for the clinical application of therapeutic OncoFAP derivatives such as [^177^Lu]Lu-DOTAGA-OncoFAP, retaining selective uptake in the FAP-positive tumor already 10 min after injection (32% injected dose [ID]/g), with a sustained uptake (i.e., higher than 20% ID/g) in the lesions over the first 6 h time window and the more recently developed [^177^Lu]Lu-DOTAGA-BiOncoFAP, exhibiting a more stable and prolonged tumor uptake than [^177^Lu]Lu-DOTAGA-OncoFAP (~20% ID/g vs. ~4% ID/g, at 24 h p.i., respectively) favorable tumor-to-organ ratios with low kidney uptake as well potent anti-tumor efficacy when administered at therapeutic doses in tumor-bearing mice [25].

All OncoFAP derivatives based on Gallium-68 and Aluminum-fluorine-18 studied in this work show excellent stability without the need of a stabilizer if not the small percentage of ethanol found in the preparation coming from purification process. On the contrary [^177^Lu]Lu-DOTAGA-OncoFAP- shows an intrinsic instability when there are high radioactive concentrations, therefore necessitating the addition of gentisic acid as a radical scavenger normally formed due to the β^−^ decay of Lutetium-177. All the OncoFAP derivatives also retained high affinity towards hFAP as shown from the coelution experiment of [^68^Ga] Ga-DOTAGA-OncoFAP with hFAP and by the cell binding experiments on FAP-positive SK-RC-52 cells. It is worth noticing that the cell binding experiments demonstrated a large difference in binding values among the compounds, varying from 0.25% for [^18^F]AlF-NOTA-OncoFAP to 2.5% for [^68^Ga]Ga-DOTAGA-OncoFAP and 22% for [^177^Lu]Lu-DOTAGA-OncoFAP. We attributed this large difference as a consequence of the difference in the molar activity in the different preparations [43]. The excellent binding proprieties of OncoFAP derivatives were confirmed by the in vivo biodistribution studies in tumor-bearing mice, which confirm the favorable tumor-targeting performance of [^18^F]AlF-NOTA-OncoFAP, comparable to that reported for [^68^Ga]/[^177^Lu]-DOTAGA-OncoFAP [8,25]. [^18^F]AlF-NOTA-OncoFAP selectively accumulates in solid lesion, with a high tumor-to-background ratio at early time points (i.e., 1 h after systemic administration, the biodistribution time normally used for PET images), thus supporting further development of the molecule as diagnostic PET radiopharmaceutical in clinical practice. For comparison, [^68^Ga]Ga-DOTAGA-OncoFAP biodistribution demonstrated beneficial tracer kinetics and high uptake in murine FAP-expressing tumor models with high tumor-to-blood ratios of 8.6 ± 5.1 at 1 h and 38.1 ± 33.1 at 3 h p.i. [8] Application of [^18^F]AlF-NOTA-OncoFAP as a PET/CT imaging agent may result in significant logistical and clinical advantages as a consequence of the imaging characteristic of the isotope. Fluorine-18 is characterized by ~97% β^+^ emission, 635 keV maximum positron energy, short β^+^ trajectory with mean positron range of 0.27 mm in soft tissue. Moreover, it can be produced in large scale in cyclotrons [44,45] with the possibility of a subsequent delivery to satellite sites other than the production site, as the half-life of the isotope is long enough to allow this strategy (t_1/2_ = 109.8 min). Translation in the clinical setting will be of significant impact also in terms of daily organization of the PET/CT schedule. In particular, the use of fluorine-18-based FAPI-derivatives allow an easier patient preparation (no need of fasting and maintaining low glucose level respect the use of FDG) and offer a longer imaging time window of approximately 30–180 min after tracer injection as compared to gallium-68 FAPI derivatives. In addition, OncoFAP-derivatives can be exploited as PET/CT imaging agents for a broader window of malignancies, which are not efficiently detected by other marketed radiopharmaceuticals, such as [^18^F]FDG, [^18^F]FLT, [^18^F]F-MCH and [^18^F]F-PSMA-1007 [1,46]. Examples are represented by oesophageal [47], liver [48] and pancreatic cancers [4], brain primary tumors and metastases [49] or head and neck cancers [50]. However, the results from the synthesis using the Aluminum-Fluorine-18 approach still warrant further development and optimization before the method can be efficiently translated into clinical use. In fact, progress is needed to increase the labeling yield and molar activities while minimizing the precursor, thus avoiding a post-radiolabeling purification step to obtain a product with high molar activity.

Our results provide the reference for a robust and efficient radiolabeling method for OncoFAP-radioligand products that are easy to implement in clinical practice in small-, medium- and large-sized hospital radiopharmacy. The different radiosynthetic strategies presented in this article for each radioisotope can be chosen on the basis of the specific needs of the various clinical centers. Automated production of [^68^Ga]Ga-DOTAGA-OncoFAP, a clinically validated PET tracer [8], and of [^177^Lu]Lu-DOTAGA-OncoFAP will facilitate the implementation of this theranostic pair in clinical practice. Efficient production of [^18^F]AlF-NOTA-OncoFAP will represent the basis for the implementation of this novel radiotracer for large-scale PET imaging applications.

## 4. Materials and Methods

### 4.1. Chemical Synthesis

Detailed chemical procedures and compound characterization for the synthesis of NOTA-OncoFAP and NODAGA-OncoFAP are reported in the Appendix A. DOTAGA-OncoFAP has been produced as previously described [9].

### 4.2. Radiochemistry

#### 4.2.1. [^68^Ga]GaOncoFAP-Derivatives Synthesis

[^68^Ga]Ga-DOTAGA/NODAGA/NOTA-OncoFAP (15–25 µg) were synthesized in an FASTlab synthesis module (GE Healthcare); the configuration of the cassette is presented in the Appendix A. The synthesis was carried out using ^68^Ga (t_1/2_ = 68 min, β^+^ = 89%, and EC = 11%) automatically eluted with 0.1 M HCl (4.5 mL, TRASIS ALLinONE reagent kit) from a 1.85 GBq (50 mCi) ^68^Ge/^68^Ga radionuclide generator (Eckert & Ziegler 1850 MBq, GalliaPharm, Radiopharma GmbH) without pre-purification. About 3.5 mL of the [^68^Ga]GaCl_3_ eluate and the precursor, dissolved in sodium acetate (750 μL, 0.7 M TRASIS ALLinONE reagent kit), were transferred into the reaction vessel. After stirring for 5 min at 95 °C under gentle nitrogen flow, the reaction mixture was loaded for purification onto the preactivated C18 cartridge (Waters, Milford, MA, USA), washed with 0.9% NaCl solution (5 mL, TRASIS ALLinONE reagent kit) and eluted with a mixture of 700 μL of absolute ethanol (TRASIS ALLinONE reagent kit) and 800 μL of water for injection (WFI, Fresenius kabi). Then, the product was diluted with 0.9% NaCl solution to obtain the final formulation obtaining a volume of 12 mL.

For the synthesis of [^68^Ga]Ga-DOTAGA-OncoFAP and [^68^Ga]Ga-NODAGA-OncoFAP, a simple preformulated radiopharmaceutical kit (OncoFAP-kit) was developed. This novel kit is ‘ready-to-use’ for radiolabeling with ^68^Ga eluted by commercially available ^68^Ge/^68^Ga generators, as in the SomaKit TOC [32]. To this aim, we develop an approach consisting of generator elution with HCl (5 mL, 0.1 M TRASIS ALLinONE reagent kit) directly in the vial containing the precursor; at the end of the elution either sodium acetate (900 µL, 0.7 M TRASIS ALLinONE reagent kit) to have a reaction pH of 4.5 or sodium formate (600 μL, 1 M BioULTRA Formic acid solution 1.0 M in H_2_O, Sigma Aldrich Taufkirchen, Germany and Sodium hydroxide solution BioUltra, for molecular biology, 10 M in H_2_O) to have a reaction pH of 3 is added. Once the generator has been eluted into the vial containing the precursor and the appropriate buffer has been added, the vial is then placed in a thermoblock preheated to 98 °C and heated for 10 min. The synthesis was repeated in different conditions, varying the amount of precursor (10–40 µg) and the reaction pH (3.2–4.5).

#### 4.2.2. [^18^F]AlF-OncoFAP

Radiolabeling of NODAGA-OncoFAP and NOTA-OncoFAP (200–300 µg) with ^18^F was performed via aluminum (Al^3+^) [^18^F]fluoride complex. In addition, for this synthesis, we used the FASTlab synthesis module. The configuration of the cassette is detailed in Appendix A.

Fluorine-18 was transferred to the module and trapped on a preactivated Sep-Pak light Accel plus QMA cartridge (Cl^−^ form: Waters, Milford, MA, USA). The cartridge was washed with 6 mL of water (HPCE grade, Sigma Aldrich) and subsequently eluted with 500 μL of the eluent solution, composed of 250 μL of 0.9% NaCl solution (99.999% trace metals basis NaCl, Sigma Aldrich) in water for injection (WFI, Fresenius kabi) and 250 μL absolute ethanol (TRASIS ALLinONE reagent kit), into a 5 mL reactor vial prefilled with 25 μL of 2 mM aluminum chloride solution (AlCl_3_, anhydrous, powder, 99.999% trace metals basis, Sigma-Aldrich) in sodium acetate buffer (0.1 M, pH 4.1). After stirring for 5 min at room temperature under gentle nitrogen flow to allow the formation of [^18^F]AlF, the solution of the precursor (600 μL of 350 μg/mL NOTA-OncoFAP or NODAGA-OncoFAP in sodium acetate 0.1 M pH 4.5) was added to the reactor vial, which was sealed and heated for 10 min at 95 °C. Next, the mixture was cooled to 40 °C. The reaction was diluted with 3.5 mL of 0.9% NaCl solution, loaded for purification onto the preactivated C18 cartridge (Waters, Milford, MA, USA), washed with 5 mL of 0.9% NaCl solution and eluted with 1.5 mL of absolute EtOH (TRASIS ALLinONE reagent kit). The product is diluted with 0.9% NaCl solution to obtain the final formulation obtaining a final volume of 12 mL. A method using semi-preparative HPLC purification is given in the Appendix A.

#### 4.2.3. [^177^Lu]Lu-OncoFAP

[^177^Lu]Lu-DOTAGA-OncoFAP was synthesized in an FASTlab synthesis module (GE Healthcare), at different molar activities ranging from 44 to 105 GBq/µmol. The configuration of the cassette is presented in Appendix A. Carrier-free Lutetium-177 (EndolucinBeta^®^ 40 GBq/mL—pharmaceutical precursor, solution) with a concentration activity of 37 MBq/µL in a volume of 0.5 mL acetate buffer at pH 4.5 is transferred to the reactor vial. The reaction was scaled down to an amount of DOTAGA-OncoFAP ranging from 5.1 to 13.3 µg, which was dissolved in acetate buffer (1 mL, 0.1 M pH 4.5 TRASIS ALLinONE reagent kit) and aspirated into the reactor vessel. The solution was stirred for 30 min at 90 °C under gentle nitrogen flow, then loaded for purification onto the preactivated C18 cartridge, washed with 5 mL of 0.9% NaCl solution and eluted with eluent solution made of 700 µL of absolute ethanol (TRASIS ALLinONE reagent kit) and 800 µL of water for injection (WFI, Fresenius kabi). The product was diluted with 0.9% NaCl solution to obtain the final formulation. Further, the radiolabeling was also performed by adding 20 mg of gentisic acid [16] to assess its effects on the radiolytic stability as radical scavenger.

#### 4.2.4. Reproducibility of the Method

To test the robustness of the synthesis method, we varied the reaction time and the amount of precursor used in the tests. We repeated each experiment at least three times to test the reproducibility.

### 4.3. Quality Controls

Radiochemical, chemical and radionuclide purity, pH, half-life, residual organic solvents, filter integrity and endotoxin content were assessed for all the radiolabeled preparations of OncoFAP derivatives.

Radiochemical and chemical purity were determined by HPLC (Berthold HERM LB500 radio detector with Jasco MD-2010 DAD at 250 nm) using a RP-18 column (Phenomenex Luna^®^ column 150 × 3 mm 3 µm 100 Å) with a flow of 0.6 mL/min at 40 °C. As mobile phase, water + TFA 0.1% (*v*/*v*, phase A) and acetonitrile + TFA 0.1% (*v*/*v*, phase B) were used with a gradient 0 to 0.5 min 90% eluent A, 0.5 to 10 min linear gradient elution from 90% to 0% eluent A, 10 to 14.5 min 0% eluent A, 14.50 to 15 min from 0% to 90% eluent A. The OncoFAP derivatives complex with the non-radioactive isotopes (Gallium-69, Fluorine-19 and Lutetium-175) were used as reference standards. iTLC on silica gel using citrate as a buffer eluent (0.1 M, pH 5) was used to quantify gallium-68 hydroxide with (Rf of 0–0.1), the labeled product (Rf between 0.3 and 0.5) and free gallium-68 (Rf between 0.6 and 0.8) both for the automated synthesis and the simple kit formulation.

For the determination of radionuclide purity, a sample of final product with a known activity and volume was analyzed (energetic spectrum from 0 to 1800 KeV) after one day in case of ^68^Ga-compounds (^68^Ga T_decay_ > 20 half-lives) and 1.5 days for [^18^F]AlF-OncoFAP derivatives (^18^F T_decay_ > 20 half-lives). To this aim, we selected only radioactive impurities with a half-life longer than 2 h. The pH of the final product was measured with pH indicator strips (pH range 2.0 to 9.0, MColorpHast™, Merck, Milano, Italy). Final activity of ^68^Ga/Al^18^F-OncoFAP was measured with an appropriate interval between the three measurements of 5 min. Residual organic solvents were measured using a gas chromatography system with a macrogol 20,000 column (30 m × 0.53 mm × 1 μm, Varian, Santa Clara, CA, U.S.) following the pharmacopoeia 2.2.28 (inlet 140 °C temperature, isocratic column temperature 50 °C, flame ionization detection temperature 250 °C during 5 min). Retention time of 2.9 min for ethanol and 4.75 min for *n*-propanol were used as reference standard.

Bubble point test of the Millex-GS vented filter was performed closing the vented portion of the filter and gradually pushing down the plunger inside a syringe filled with air to increase the pressure on the pressure gauge. The pressure when a continuous stream of air bubbles that appeared out of the 0.22 μm membrane filter was measured as the product-wetted bubble point value.

Endotoxin determination was conducted with a Endosafe PTS Reader^®^ rapid test (Charles River).

### 4.4. In Vitro Stability and Lipophilicity

The in vitro stability of the radiopharmaceuticals preparations in saline solution was evaluated on the bulk solution i.e., with the highest radioactive concentration. In addition, the in vitro stability of radiopharmaceutical preparations was tested by incubating 10 MBq of each radiolabeled compounds in 1 mL of human plasma at 37 °C. Small aliquots of bulk solution were analyzed by HPLC and iTLC and 10 µL of plasma solution were analyzed by iTLC at 30, 60, 90, 120, 150 and 180 min for [^68^Ga]Ga-DOTAGA/NODAGA-OncoFAP derivatives, 50, 100, 150 and 200-min for [^18^F]AlF-NOTA/NODAGA-OncoFAP or 1, 3, 6, 12, 24, and 2, 3, 4, 7 and 9 days for [^177^Lu]Lu-DOTAGA-OncoFAP, as show in Figure 3.

Lipophilicity was determined by partitioning between *n*-octanol and saline solution (LogP) or PBS (LogD_7.4_) at room temperature using the conventional shake-flask method. An aliquot of the formulated solution containing ~500 kBq of radiopharmaceuticals was added to a tube containing 6 mL of *n*-octanol/saline solution or *n*-octanol/PBS (1:1 *v*/*v*, LogP or LogD_7.4_, respectively). The tubes were shaken for 20 min, followed by centrifugation (5000× *g* for 5 min) and separation of the phases. Aliquots of 1 mL of the organic and the aqueous phases were taken, and the activity was measured using an automated gamma counter. The LogP or LogD_7.4_ were calculated as the decadic logarithm of [activity (cpm/mL) in *n*-octanol]/[(activity (cpm/mL) in saline solution or PBS].

### 4.5. Characterization of OncoFAP Binding to hFAP

To assess the binding properties of radiolabeled OncoFAP-derivatives preparations to hFAP, [^68^Ga]/[^18^F]AlF/[^177^Lu]-OncoFAP derivatives (500 kBq) were incubated with hFAP (2 μM, 100 μL) and then loaded on PD-10 columns, pre-equilibrated with running buffer (50 mM Tris, 100 mM NaCl, and 1 mM ethylenediaminetetraacetic acid [EDTA], pH = 7.4) and eluted with running buffer. Fractions of the flow through (500 μL) were collected, and the radioactivity intensity was measured immediately on a gamma counter (λ = 511 KeV for gallium-68 and fluorine-18 and λ = 208 KeV for lutetium-177) [9].

### 4.6. Cell Binding

Cell binding assays were performed in triplicate on SK-RC-52.hFAP and SK-R-RC-52.wt cells (density of 1.5 × 10^5^ cells each tube), growth in RPMI-1640 medium (Gibco) containing 10% Fetal Bovine Serum (Gibco) and 1% Antibiotic-Antimycotic (Gibco) at 37 °C in 5% carbon dioxide (CO_2_). After adding [^68^Ga]Ga-DOTAGA-OncoFAP, [^177^Lu]Lu-DOTAGA-OncoFAP or [^18^F]AlF-NOTA-OncoFAP (10 µL, 0.1 MBq) to the culture medium, the cells were incubated at 37 °C for 10, 30, 60 min for ^68^Ga-derivatives and 10, 30, 60, 120, 240 min for ^18^F-derivatives or 10, 30, 60, 120, 240, 360 min and 24 and 48 h for the ^177^Lu-derivative. As a control, nonspecific binding determination was carried out by incubating with the OncoFAP derivatives complex with the non-radioactive isotopes reference standard (100 µM) with the same cell lines for 5 min before incubation with the radiolabeled compounds. After incubation, the cell pellets were washed twice with 1 mL of cold PBS (pH 7.4), centrifuged (400× *g*, 5 min) and the pellet measured by an automatic gamma counter (WIZARD2, PerkinElmer, Waltham, MA, USA).

### 4.7. Animal Studies

#### 4.7.1. Implantation of Subcutaneous Tumors

Tumor cells were grown to 80% confluence in Dulbecco’s Modified Eagle Medium (DMEM) supplemented with 10% Fetal Bovine Serum (Thermofisher, Waltham, MA, USA) and 1% antibiotic-antimycotic (Gibco) and detached with Trypsin-EDTA 0.05%. HT-1080.hFAP, cells (FAP positive cells) were resuspended in Hanks’ Balanced Salt Solution medium. Aliquots of 5 × 10^6^ cells (100 μL of suspension) were injected subcutaneously in the right flanks of female athymic Balb/c AnNRj-Foxn1 mice (6 to 8 weeks of age, Janvier-labs, Le Genest-Saint-Isle, France).

#### 4.7.2. Biodistribution Studies in Tumor Bearing Mice

Female athymic Balb/c AnNRj-Foxn1 mice (6 to 8 weeks of age) were implanted in the right flank with HT-1080.hFAP tumors as described above. Tumors were allowed to grow to an average volume of 250 mm^3^. Mice were randomized (n = 5 per group) and injected intravenously with radiolabeled preparations of [^18^F]AlF-NOTA-OncoFAP (10 nmol/mice; 77 kBq; RCP:89%). Mice were euthanized 1 h after the injection by CO_2_ asphyxiation. Organs were extracted, weighted, and radioactivity was measured with a Packard Cobra Gamma Counter. Values are expressed as percent ID/g ± SD. We did not repeat the biodistribution experiments using [^68^Ga]Ga-DOTAGA-OncoFAP and [^177^Lu]Lu-DOTAGA-OncoFAP since they are already reported [8,9].

## 5. Conclusions

Efficient production of pharmaceutical-grade [^68^Ga]Ga-OncoFAP and [^18^F]AlF-OncoFAP can be achieved through automatic radiolabeling procedures described in this article. [^68^Ga]Ga-OncoFAP can be obtained with a simple kit-based approach, thus allowing small radiopharmacies to offer this new PET radiotracer to cancer patients with limited financial burden. Furthermore, we have established methodologies to efficiently produce an OncoFAP-based radiotheranostic pair in which [^68^Ga]Ga-DOTAGA-OncoFAP is used as a companion diagnostic of the targeted therapeutic agent [^177^Lu]Lu-DOTAGA-OncoFAP. The procedures described in this article are being implemented in the clinical protocols of theranostic trials with OncoFAP-derivatives to enable imaging and therapy of patients with solid tumors.

## Figures and Tables

**Figure 1 pharmaceuticals-15-00958-f001:**
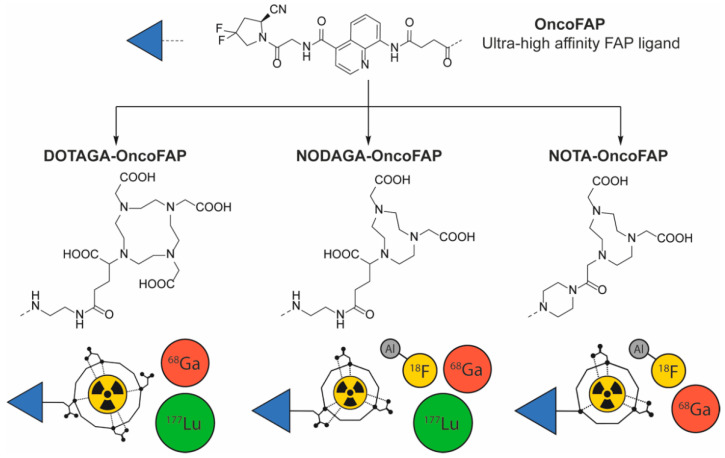
Chemical structures of OncoFAP and its conjugates with radiometal chelators. OncoFAP is a portable small organic ligand that targets Fibroblast Activation Protein in solid tumors with ultra-high affinity. Structures of OncoFAP and its DOTAGA, NODAGA and NOTA conjugates are presented. DOTAGA-OncoFAP conjugate was radiolabeled with gallium-68 and lutetium-177. NODAGA-OncoFAP conjugate was radiolabeled with gallium-68, lutetium-177 and fluorine-18. NOTA-OncoFAP- derivative was radiolabeled with gallium-68 and fluorine-18.

**Figure 2 pharmaceuticals-15-00958-f002:**
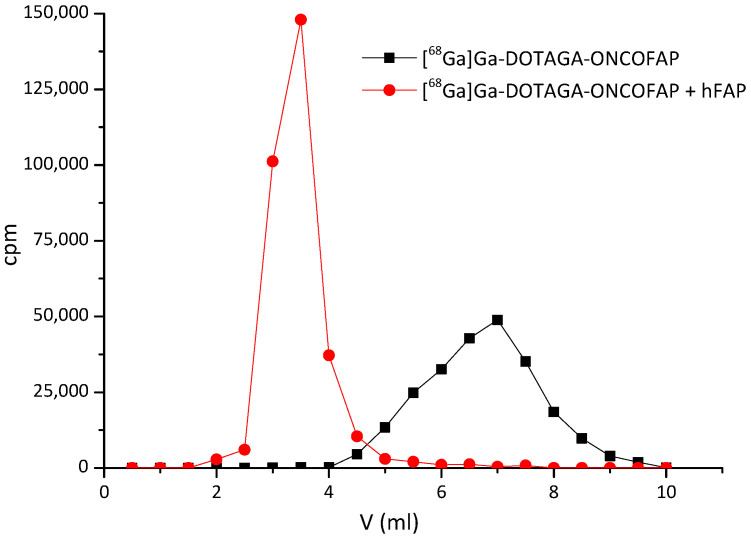
Co-elution experiments performed with [^68^Ga]Ga-DOTAGA-OncoFAP conjugate. [^68^Ga]Ga-DOTAGA-OncoFAP forms a stable complex with recombinant hFAP. [^68^Ga]Ga-DOTAGA-OncoFAP (500 kBq) and hFAP (2 μM, 100 μL) eluted on a PD-10 column.

**Figure 3 pharmaceuticals-15-00958-f003:**
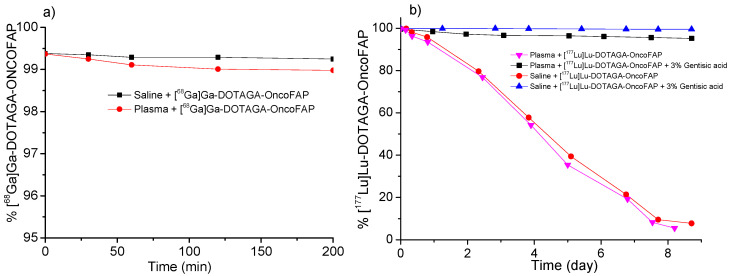
In vitro radiopharmaceuticals’ stability in saline or human plasma at 37 °C. Small aliquots were analyzed by HPLC (**a**) 30–60–90–120–150–180 min for [^68^Ga]Ga-DOTAGA-OncoFAP and (**b**) 1–3–6–12–24–48–72–96 h and 7 days for [^177^Lu]Lu-DOTAGA-OncoFAP.

**Figure 4 pharmaceuticals-15-00958-f004:**
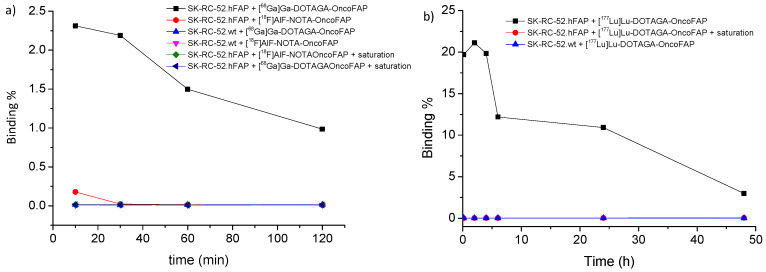
Cell binding assays in SK-RC-52.hFAP and SK-R-RC-52 wild-type (SK-RC-52.wt) cells (density of 1.5 × 10^5^ cells each tube). (**a**) [^68^Ga]Ga-DOTAGA-OncoFAP and [^18^F]AlF-NOTA-OncoFAP (10 µL, 0.1 MBq) cells were incubated at 37 °C for 10, 30, 60 and 120 min. (**b**) [^177^Lu]Lu-DOTAGA-OncoFAP (10 µL, 0.1 MBq) cells were incubated at 37 °C for 10, 120, 240, 360 min and 24 and 48 h.

**Figure 5 pharmaceuticals-15-00958-f005:**
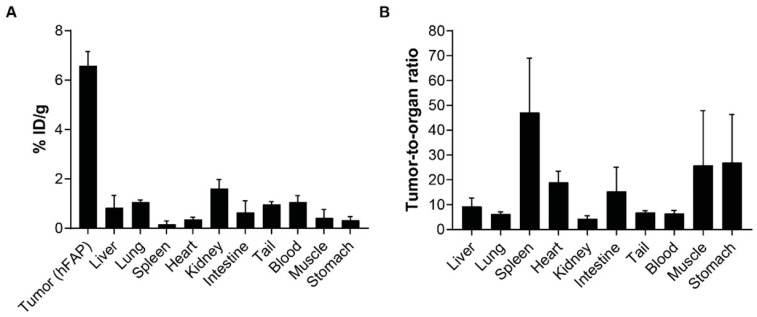
(**A**) Quantitative in vivo biodistribution and (**B**) tumor-to-organ ratios of [^18^F]AlF-NOTA-OncoFAP in nude mice bearing subcutaneous HT-1080.hFAP tumors. [^18^F]AlF-NOTA-OncoFAP was intravenously injected at 500 nmol/Kg dose (3.85 MBq/Kg), animals were sacrificed 1-h post-administration and distribution of the radiopharmaceutical was assessed by gamma-counter.

**Table 1 pharmaceuticals-15-00958-t001:** Different synthetic conditions used for [^68^Ga]Ga-NOTA-OncoFAP.

Radiosynthesis [^68^Ga]Ga-NOTA-OncoFAP			
	Condition 1	Condition 2	Condition 3
NOTA-OncoFAP (µg)	15	20	25
Reaction time (min)	10	5	5
Starting activity (MBq)	715 ± 115	780 ± 150	770 ± 135
Final activity (MBq)	580 ± 110	641 ± 135	650 ± 116
RCY (%)	78 ± 3	82 ± 2	84 ± 1
RCY decay corrected (%)	89 ± 3	88 ± 2	90 ± 1
Final molar activity (GBq/µmol)	39 ± 7	32 ± 7	26 ± 5
RCP (%)	99.1 ± 0.1	99.1 ± 0.1	99.1 ± 0.1

**Table 2 pharmaceuticals-15-00958-t002:** Different synthetic conditions used for [^68^Ga]Ga-NODAGA-OncoFAP.

Radiosynthesis [^68^Ga]Ga-NODAGA-OncoFAP			
	Condition 1	Condition 2	Condition 3
NODAGA-OncoFAP (µg)	15	20	25
Reaction time (min)	10	5	5
Starting activity (MBq)	1085 ± 95	1000 ± 74	1100 ± 60
Final activity (MBq)	880 ± 70	840 ± 56	954 ± 28
RCY (%)	81 ± 2	84 ± 2	87 ± 2
RCY decay corrected (%)	93 ± 2	90 ± 2	93 ± 3
Final molar activity (GBq/µmol)	59 ± 5	42 ± 3	38 ± 2
RCP (%)	99.2 ± 0.1	99.2 ± 0.1	99.2 ± 0.1

**Table 3 pharmaceuticals-15-00958-t003:** Different synthetic conditions used for [^68^Ga]Ga-DOTAGA-OncoFAP.

Radiosynthesis [^68^Ga]Ga-DOTAGA-OncoFAP			
	Condition 1	Condition 2	Condition 3
DOTAGA-OncoFAP (µg)	15	20	25
Reaction time (min)	10	5	5
Starting activity (MBq)	1159 ± 93	1208 ± 93	1159 ± 57
Final activity (MBq)	924 ± 99	942 ± 60	915 ± 33
RCY (%)	75 ± 3	78 ± 2	79 ± 2
RCY decay corrected (%)	84.0 ± 3	87 ± 2	88 ± 2
Final molar activity (GBq/µmol)	55 ± 6	42 ± 3	33 ± 1
RCP (%)	99.3 ± 0.1	99.3 ± 0.1	99.4 ± 0.1

**Table 4 pharmaceuticals-15-00958-t004:** Acceptance criteria and average result of five [^68^Ga]-derivatives synthesis obtained with condition of condition 3.

Parameters	Acceptance Criteria	Average NOTA-OncoFAP	Average NODAGA-OncoFAP	Average DOTAGA-OncoFAP
[^68^Ga]-labeled activity MBq	200–1300	520–580	900–970	850–920
Volume (mL)	12	12	12	12
Aspect	Clear colorless solution	Clear colorless solution	Clear colorless solution	Clear colorless solution
Half-life (min)	62–74	66.8	66.9	67.1
- ^68^Ga^3+^ - ^68^Ga colloidal - RCP	≤2% ≤3% ≥95%	0.1 ± 0.1% 0.8 ± 0.1% 99.1 ± 0.1%	0.1 ± 0.1% 0.7 ± 0.1% 99.2 ± 0.1%	0.1 ± 0.1% 0.5 ± 0.1% 99.4 ± 0.1%
Ethanol concentration (*v*/*v*)	<10%	4.9% ± 0.5%	4.8% ± 0.5%	4.9% ± 0.5%
^68^Ge breakthrough	<10^−3^%	<10^−4^%	<10^−4^%	<10^−4^%
pH	4.0–8.0	5.0 ± 0.3	5.0 ± 0.3	5.0 ± 0.3
Bacterial endotoxins	<175 EU/V	<6 EU/V	<6 EU/V	<6 EU/V
Filter integrity (psi)	≥50	>50	>50	>50

**Table 5 pharmaceuticals-15-00958-t005:** Average value obtained for the synthesis of [^68^Ga]Ga-NODAGA-OncoFAP.

Radiosynthesis [^68^Ga]Ga-NODAGA-OncoFAP (Simple Preparation)		
	Formate Buffer	Acetate Buffer
NODAGA-OncoFAP (µg)	40	40
Reaction pH	3.0 ± 0.1	4.4 ± 0.1
Starting activity (GBq)	1.1 ± 0.2	1.1 ± 0.2
Final molar activity (GBq/µmol)	26 ± 5	26 ± 5
- ^68^Ga^3+^ content - Colloidal ^68^Ga content - RCP	3.4 ± 2.6% 1.5 ± 0.2% 95.1 ± 2.5%	7.9 ± 1.7% 4.6 ± 1.7% 87.5 ± 2.8%

**Table 6 pharmaceuticals-15-00958-t006:** Average value obtained for the synthesis of [^68^Ga]Ga-DOTAGA-OncoFAP.

Radiosynthesis [^68^Ga]Ga-DOTAGA-OncoFAP (Simple Preparation)		
	Formate Buffer	Acetate Buffer
DOTAGA-OncoFAP (µg)	40	40
Reaction pH	3.0 ± 0.1	4.4 ± 0.1
Starting activity (GBq)	1.1 ± 0.2	1.1 ± 0.2
Final molar activity (GBq/µmol)	26 ± 5	26 ± 5
- ^68^Ga^3+^ content - Colloidal ^68^Ga content - RCP	3.1 ± 1.1% 2.9 ± 0.3% 94.0 ± 1.4%	5.6 ± 2.7% 6.4 ± 1.8% 88.0 ± 2.4%

**Table 7 pharmaceuticals-15-00958-t007:** Different conditions used for the synthesis of [^18^F]AlF-NOTA-OncoFAP.

Radiosynthesis of [^18^F]AlF-NOTA-OncoFAP			
	Condition 1	Condition 2	Condition 3
NOTA-OncoFAP (µg)	200	250	300
Initial activity (MBq)	3970 ± 203	9250 ± 147	16280 ± 976
Product activity (MBq)	678 ± 33	1750 ± 100	2967 ± 25
RCY (%)	16.9 ± 0.5	18.9 ± 0.3	18.2 ± 1.2
RCY decay corrected (%)	18.7 ± 0.5	20.9 ± 0.3	20.2 ± 1.2
Final molar activity (GBq/µmol)	3.05 ± 0.3	6.6 ± 0.8	9.3 ± 0.7
RCP (%)	92.0 ± 0.5	89.6 ± 0.8	90.5 ± 0.9

**Table 8 pharmaceuticals-15-00958-t008:** Different conditions used for the synthesis of [^18^F]AlF-NODAGA-OncoFAP.

Radiosynthesis of [^18^F]AlF-NODAGA-OncoFAP			
	Condition 1	Condition 2	Condition 3
NODAGA-OncoFAP (µg)	200	250	300
Initial activity (MBq)	3219 ± 150	4255 ± 185	6660 ± 220
Product activity (MBq)	78 ± 15	96 ± 21	137 ± 15
RCY (%)	2.4 ± 0.5	2.3 ± 0.5	2.1 ± 0.3
RCY decay corrected (%)	2.6 ± 0.5	2.5 ± 0.5	2.3 ± 0.3
Final molar activity (GBq/µmol)	0.36 ± 0.10	0.35 ± 0.10	0.42 ± 0.13
RCP (%)	60.3 ± 5.2	68.4 ± 5.8	67.5 ± 4.8

**Table 9 pharmaceuticals-15-00958-t009:** Acceptance criteria and average result of five [^18^F]AlF-OncoFAP synthesis obtained with condition of batch 3.

Parameters	Acceptance Criteria	Average NOTA-OncoFAP	Average NODAGA-OncoFAP
[^18^F]-labeled activity	200–4000 MBq	2806–3205 MBq	210–230 MBq
Aspect	Clear colorless solution	Clear colorless solution	Clear colorless solution
RCY (%)	95%	20.2 ± 1.5	2.0 ± 0.7
- ^18^F^-^content - Other - RCP	≤2% ≤3% ≥95%	4.1 ± 0.2% 5.9 ± 0.4% 90.6 ± 0.6%	12.4 ± 2.5% 27.3 ± 1.4% 60.3 ± 1.3%
Ethanol concentration (*v*/*v*)	<10%	5.6 ± 0.5%	5.4 ± 0.5%
pH	4.0–8.0	5.5 ± 0.3	5.5 ± 0.3
Bacterial endotoxins	<175 EU/V	<6 EU/V	<6 EU/V
Filter integrity (psi)	≥50	>50	>50

**Table 10 pharmaceuticals-15-00958-t010:** Different conditions used for the synthesis of [^177^Lu]Lu-DOTAGA-OncoFAP.

Radiosynthesis of [^177^Lu]Lu-DOTAGA-OncoFAP				
	Condition 1	Condition 2	Condition 3	Condition 4
DOTAGA-OncoFAP (µg)	13.3 ± 0.25	5.1 ± 0.15	8.2 ± 0.50	5.3 ± 0.15
Reaction volume (mL)	2.5	2.5	2.5	1.5
Initial activity (MBq)	592 ± 9	870 ± 6	1047 ± 19	573 ± 6
Product activity (MBq)	577 ± 8	307 ± 5	670 ± 25	533 ± 6
RCY %	97.4 ± 0.4	35.3 ± 1.9	54.8 ± 1.3	92.9 ± 0.75
Gentisic acid (mg)	0	20	20	20
Final molar activity (GBq/µmol)	43 ± 2	60 ± 5	83 ± 3	100 ± 3
RCP (%)	> 99.8	>99.8	> 99.8	> 99.8

**Table 11 pharmaceuticals-15-00958-t011:** Acceptance criteria and average result of three synthesis of [^177^Lu]Lu-DOTAGA-OncoFAP condition 4.

[^177^Lu]Lu-DOTAGA-OncoFAP		
Parameters	Acceptance Criteria	Average DOTAGA-OncoFAP
[^177^Lu]-labeled activity (GBq)	5–9	7.4–7.7
Aspect	Clear colorless solution	Clear colorless solution
- ^177^LuCl_3_ - Others - [^177^Lu]Lu-DOTAGA-OncoFAP-	≤2% ≤3% ≥95%	1.9 ± 0.5% - 98.1 ± 0.5%
Ethanol concentration (v/v)	<10%	3.2 ± 0.9%
pH	4.0–8.0	5.5 ± 0.3
RCY		95 ± 2.6%
Bacterial endotoxins	<175 EU/V	<6 EU/V
Final molar activity (GBq/µmol)		105 ± 5
Filter integrity (psi)	≥50	>50

## Data Availability

Data is contained within the article and Appendix A.

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
