# Peer review of "Automated Radiosynthesis, Preliminary In Vitro/In Vivo Characterization of OncoFAP-Based Radiopharmaceuticals for Cancer Imaging and Therapy"

_pharmaceuticals, 2022, doi:10.3390/ph15080958_

Round 1

Reviewer 1 Report

The paper by Bartoli et al. reports on labelling procedures of Onco-FAP based radiopharmaceuticals. The topic is of current interest, given the raising impact of FAP-based imaging and radionuclidic therapy, and reporting on streamlined pharmaceutical manufacturing processes is timely.

However, I feel that there is a substantial mismatch between the title of the paper and its conclusion, and the rest of the manuscript text. In fact, while title and conclusions are centered on optimizing automated processes of synthesis of such important radiopharmaceuticals, the manuscript text has extensive sections regarding in vitro and ex vivo characterization that falls behind the scope of the manuscript. A lot of useful information regarding the process, such as precursor synthesis and synthesizer schemes, is relegated in the supporting info.

If the authors are willing to keep the scope of their paper as claimed in the title (choice I would support, as the pharmacological properties of these tracers is not in discussion), they need to move the pharmaceutical characterization (basically all section 2.2, 4.5, 4.6, 4.7 and related material) in the supporting info section and move the synthetic info (nearly the entire current supp info document) in the main text. In doing that, they should clearly state processes times as well as yields, and select meaningful chromatographic profiles to demonstrate the quality of the product, especially comparing kit and automated synthesis for 68Ga products.

If the authors are willing to keep the in vivo and ex vivo data , they should completely change the focus of the paper towards a theranostics point of view, and add a substantial amount of data, such as: in vivo PET imaging of various tracers, proof of 177Lu treatment by ex vivo assessment and by in vivo imaging.

In any case, the submitted text can be further improved following the comments below:

L83 and following: the compounds numbers are cited, but those numbers are not present in any figure.

l94: the referring Tables are placed much later in the manuscript, and this makes reading difficult. The same happens for Figures, that are not properly placed to make the text easily referenceable.

l94: "radiolabeling yield" is not the terminology suggested in the Society of Radiopharmaceuticals Sciences nomenclature guidelines; Radiochemical Yield or Activity Yield should be used, in the acceptation agreed. The same applies to the Tables.

l95-97: the argument on precursor amount is unclear; the claimed best is 25ug, but increasing to 20ug and 25ug did not provide differences? Maybe the claimed best should be 15ug?

l99: how were the products purified? State if no purification was performed.

l108: use always RCP (once it has been defined as "radiochemical purity").

l111: does this process include also HPLC purification? If so, what was the total process time?

l114: "...with RCY..." seems an incomplete statement.

l116: "the validation..." seems a new period, but there is not any punctuation.

Section 2.2 is made up of few very succinct subsections; it could make more sense to join them together, or to expand the text with more details.

l131: "...for over 2 hours..." should be just "for 2 hours", or the highest timepoint tested.

l133: "...partial radiolysis..." should be precisely specified in numbers, as well the "prevention" of it by adding gentisic acid.

l135: "... high hydrophilicity" determined how?

l143: what is "SK-RC-52.wt"? (reference in too distant figure?)

Fig1: why the "Al-F" graphics is present only on the rightmost sketch?

Table 10 and similar tables: how was the molar (not specific) activity measured?

l270: "alluminium" should be "aluminum" in all the instances

l278: Space missing at end of sentence. BTW, can the authors understand whether the radiolabelling is not efficient because the Al-F is not complexed or because the formed Al-F complex loses free fluoride in the reaction conditions? Would it be possible to assess this?

l305: should be "worth noticing"

l310: how does it compare with the actual molar activities? Are they really in the ratio ~1/10/100 like the cell binding %?

l311: "in vivo" is not completely italicized.

l328-330: the (unclear) fasting and low glucose argument is not related to the nature of 18F.

l356: superscript 68 is missing.

l370: what was the final volume of the formulation?

l374: should be "eluted"

l385: is there any purification needed in the kit process? Or at least a sterilization by filtration? Any aseptic techniques needed?

l386: superscript 18 missing.

l400: "formation of AlF" is chemically obscure; is it a +2 cation or something else?

l423: should it be "radiolytic stability"?

l439: "Cold labelled derivatives" is not a term suggested in SRS nomenclature guidelines.

l445: should be "compounds".

l463 and 465: what is the difference between stability of "preparations" (l463) and "radiopharmaceutical preparations" (l465)?

l497: using "cold" is not indicated in the SRS guidelines.

Reviewer 2 Report

This manuscript described the radiolabeling procedures for preparing 18F/68Ga/177Lu-OncoFAP agents. This study was well designed and performed, which would facilitate those agents to be used in clinical trials. Therefore, it should be published after revision in accordance with the comments below.

1. In this study, three types of OncoFAP precursors were used, but among them, NODAGA-OncoFAP and NOTA-OncoFAP seem not been reported previously. If not being reported, it should be clarified and mentioned why those precursors were developed and used in addition to DOTAGA-OncoFAP.

2. In the result section of quality control, "specific activity" was mentioned, while "molar activity" was discussed in the discussion section. That expression should be unified.

3. In the purification process of radiolabeled OncoFAP agents, only a preactivated C18 cartridge was used, which I suspected could not separate radiolabeled agents and the precursors. Therefore, it should be clarified if the precursors were purified, and if not, it should be discussed that effects on their real specific activities.

4. About bacterial endotoxins, the unit was "EU/V" in the criteria, while it was "EU/mL" in the real result. It should be unified.

5. Regarding the radiolabeling of 18F via [18F]AlF, the radiochemical yield was quite low compared with other reported [18F]AlF-labeled agents. In addition, its radiochemical yield was not meet the criteria (over 95%). Thus, these problems should be discussed.

6. The authors prepared automated synthetic methods for the "FASTlab2" module, but it seems to be used for "FASTlab", too. Thus, it should not be limited to "FASTlab2".

Reviewer 3 Report

This article offers valuable information concerning automated methods for the synthesis of OncoFAP-based probe radiolabeled with 68Ga, 18F and 177Lu. Authors tested some labeling conditions and the research itself is important for the clinical use of these radiopharmaceuticals. The manuscript is appropriate for the intent of Pharmaceuticals, however, authors should revise some points before the acceptance.

1.     The radiochemical purity of 18F-labeled OncoFAP compounds was low. Do authors have any ideas to remove 18F-impurities?

2.     How much the radiochemical purity in in vivo biodistribution study (Figure 5)? Reviewer is afraid that the accuracy might be low if the purity was not so high.

3.     Did authors check the in vivo stability of these labeled compounds?

4.     In Figure 4 a), the unit of horizontal axis would not be “h”, but “min”.

5.     Why did the cellular binding of radiopharmaceuticals decrease in the course of timepoint? Please discuss it.

Round 2

Reviewer 1 Report

Authors have properly modified the text following suggestions, except modifying the title as they proposed (Automated radiosynthesis, preliminary in vitro/in vivo characterization of OncoFAP-based radiopharmaceuticals for cancer imaging and therapy).

It is suggested for the title to be changed before publication.

Author Response

We thank the reviewer and we apologize, we have now changed the title in the manuscript as well